# MTF Measurement by Slanted-Edge Method Based on Improved Zernike Moments

**DOI:** 10.3390/s23010509

**Published:** 2023-01-02

**Authors:** Shuo Zhang, Fengyan Wang, Xiang Wu, Kangzhe Gao

**Affiliations:** College of Geo-Exploration Science and Technology, Jilin University, Changchun 130026, China

**Keywords:** MTF, slanted-edge method, Zernike moment, photogrammetry, performance evaluation

## Abstract

The modulation transfer function (MTF) is an important parameter for performance evaluation of optical imaging systems in photogrammetry and remote sensing; the slanted-edge method is one of the main methods for measuring MTF. To solve the problem of inaccurate edge detection by traditional methods under the conditions of noise and blur, this paper proposes a new method of MTF measurement with a slanted-edge method based on improved Zernike moments, which firstly introduces the Otsu algorithm to automatically determine the Zernike moment threshold for sub-pixel edge detection to precisely locate the edge points, then obtains LSF through edge point projection, ESF sampling point acquisition, smoothing, fitting, taking ESF curve differential and Gaussian fitting, and finally, accurately obtaining MTF by LSF Fourier transform and modulo normalization. Based on simulation experiments and outdoor target experiments, the reliability of the proposed algorithm is verified by the deviations of slanted-edge angle and MTF measurement, and the tolerance degree of edge detection to noise and ambiguity are analyzed. The results show that compared with ISO 12233, OMNI-sine method, Hough transform method and LSD method, this algorithm has the highest edge detection accuracy, the maximum tolerance of noise and ambiguity, and also improves the accuracy of MTF measurement.

## 1. Introduction

The modulation transfer function (MTF) is one of the important indicators for evaluating camera performance and imaging quality [1,2,3], which reflect the degree of diffusion and weakening of different spatial frequency signals in the imaging process; it is used to qualitatively and quantitatively describe the imaging capabilities of an optical imaging system [4]. General aerospace cameras are subject to strict laboratory measurements of their MTF at the factory [5], but there are differences between laboratory and actual working scenarios, and the MTF will change due to factors such as flight platform stability, changes in operating conditions, and aging and wear of parts [6]. Therefore, real-time evaluation of the MTF is conducive to better mastery of the camera performance and provides parameters for Photogrammetry and other work [7].

The acquisition of MTF can be divided into direct and indirect measurement methods. Direct measurement methods, such as the rectangular target method [8], require the production of a series of targets with different frequencies, constantly replaces the targets during the test process, and the test results are easily affected by the phase of the targets, which is not suitable for fully automated real-time testing of MTF. Indirect measurement methods include the array of point sources method [9], the edge and pulse method [10,11] and the slanted-edge method [12]. The array of point sources method and the edge and pulse method have high requirements for the brightness of the light source and the accuracy of the target. The slanted-edge method, however, only requires a contrasting and obvious slanted edge to complete the measurement, which is convenient and fast, making this the commonly used measurement method. According to the International Organization for Standardization in the ISO 12233 standard, a detailed description of the slanted-edge method (hereinafter referred to as the ISO 12233 method), which mainly uses the derivation of the gray value of the image in the row direction to determine the edge point and fit the edge, the method is computationally intensive, edge detection accuracy is general, and vulnerable to noise, blur and other factors, such as the impact of the edge. Many scholars have made improvements to address these issues: in terms of running speed, Zhu et al. [13] proposed an improved Hough transform to obtain the edge angle, which significantly improved the detection efficiency. In terms of edge detection, Li et al. [14] used an improved Canny operator that automatically determines the high and low thresholds to find the angle of edge to improve the accuracy of MTF. Yuan et al. [15] proposed a new algorithm for MTF measurement based on normal distribution fitting and median point finding to determine the edge. Qu [16] introduced the gradient operator into the edge straight-line fitting and performed high-precision sub-pixel position correction of the edge. In terms of edge selection, Qin [17] proposed a new model for calculating MTF based on arbitrary inclined edges, which solves the limitation of edge selection.

The key to measuring MTF using the slanted-edge method is to obtain the edge. The above methods have achieved good results, but they are sensitive to noise and blur. This paper proposes a new method to calculate MTF by using the Zernike moment. The Zernike moment has the characteristics of rotation invariance and good noise resistance. Combined with the Otsu algorithm, the threshold value in the Zernike moment is automatically determined, so that the sub-pixel position of the edge point is accurately located, and the edge straight line is fitted. The ESF samples obtained by projection along the edge direction are smoothed by a five-point filtering method. Then, LSF was obtained with the differential of ESF curve and Gaussian fitting, and finally, the MTF is obtained by LSF Fourier transform and modulo normalization.

In order to verify the rationality of the algorithm, a simulation experiment and outdoor target experiment are carried out. In the simulation experiment, the ideal edge image is generated by the computer and different levels of noise and blur are added to it. ISO 12233, Hough transform, LSD and the algorithm proposed in this paper are respectively used to detect the edge angle and to study the ability of each method for detecting the edge. Furthermore, on the basis of these methods, the OMNI-sine method is introduced, then MTF is measured and compared with the system MTF value simulated by the Gaussian kernel function, and the ability of measuring MTF based on the five methods is discussed. The outdoor target experiment mainly uses a ground radiation target for MTF measurement, and discusses the applicability of the proposed algorithm. The study shows that the proposed algorithm has the highest edge detection accuracy, the highest tolerance of noise and blur, and the minimum deviation between MTF and its system value at Nyquist frequency. The research improves the accuracy of the inclined edge detection and MTF measurement. 

## 2. Principle and Method of MTF Measurement

MTF is the energy response of the optical system to the sine wave with different spatial frequencies. When the imaging system images the sine wave target, the image modulation will be lower than the target modulation due to the influence of spatial resolution [18]. The image modulation *M* can be defined as:(1)$$M=\frac{A_{max}-A_{min}}{A_{max}+A_{min}}$$
where, $A_{max}$, $A_{min}$ are the maximum and minimum sine wave strength respectively. When imaging the sine wave target with frequency f, MTF can be expressed as:(2)$$\mathrm{MTF}(f)=\frac{M_{image}}{M_{object}}$$

That is the ratio of image modulation to target modulation. Because the image modulation is limited by the pixel size and focal length of the sensor, it will decrease with an increase in the spatial frequency; the MTF will also decrease with an increase in the spatial frequency. Therefore, when the spatial frequency becomes larger, the image will become more blurred.

It can be seen from the definition of MTF that the measurement of MTF can be completed by using the sine wave target, but the target production is difficult, so indirect measurement is generally used to obtain the system MTF. When the detector samples the point light sources, the two-dimensional image distribution function obtained by imaging is the point spread function (PSF) [19,20], the system MTF can be obtained by the two-dimensional Fourier transform of PSF [21].
(3)MTF(u,v)=|F[PSF(u,v)]|

The line light sources can be regarded as a collection of point light sources along a certain direction. When the point light sources are replaced by the line light sources, the two-dimensional image distribution function obtained by imaging is the line spread function (LSF); therefore, the integration of PSF along a certain direction is LSF, and the MTF perpendicular to the line direction can be obtained by the Fourier transform of LSF.
(4)$$\mathrm{LSF}(x)=\int_{-\infty }^{\infty }\mathrm{PSF}(x,y)dy$$
(5)MTF(u,0)=|F[LSF(x)]|

When imaging with an edge, the two-dimensional image distribution function obtained is the edge spread function (ESF). LSF is the differential of ESF, so the system MTF can be obtained by taking the differential of ESF and then Fourier transform, as with Formula (6).
(6)$$\mathrm{LSF}(x)=\frac{d}{dx}[\mathrm{ESF}(x)]$$

The relationship among PSF, LSF, ESF, and MTF is shown in Figure 1 below:

The production of point light sources and line light sources requires a certain precision and is relatively complex, so the method of obtaining ESF by edge imaging is generally used to calculate MTF, which is called slanted-edge method. MTF measurement of different photoelectric imaging systems can be completed with only one edge target or edge ground object meeting the reflectivity requirements; the target is easy to produce and is not affected by diffraction. The method is based on geometric projection and is simple and effective, which is convenient for engineering implementation. Firstly, ROI is selected on the original image to obtain the target edge. Secondly, ESF can be obtained by projecting the imaged pixel according to the target edge. LSF can be obtained by differentiating ESF, and finally, the LSF can be transformed into MTF by one-dimensional Fourier transform and modulo normalization. The key to the slanted-edge method is to accurately obtain the edge. The higher the accuracy of the edge, the more accurate the MTF obtained.

The steps of the slanted-edge method are described in detail in the ISO 12233 standard proposed by the International Organization for Standardization and widely used in the inspection of various aerial remote sensing cameras.

### 2.1. Traditional Method

#### 2.1.1. ISO 12233

The operation flow of the slanted-edge method is described in detail in the step of measuring and analyzing the spatial frequency response of the digital camera defined by ISO 12233 [22]. The implementation steps are as follows:Select the edge area ROI;The opto-electronic conversion function (OECF) is used to transform the image data of the region of interest to compensate the photometric response of the digital camera;Calculate the differential of the ESF in the row direction, i.e., the LSF of the system;Calculate the center position of each LSF line in the selected edge area, and fit all positions;Calculate the number of rows per phase period and adjust the ROI so that the entire ROI contains an integer number of phase periods;Project all the pixel points in the ROI to the first row of the ROI along the direction of the fitted line;Take 1/4 of the original imaging device sampling interval as the new sampling interval, calculate the geometric average of all data points falling in the same sampling interval, and use this value to represent the data information of the sampling interval, to obtain the average up sampling ESF, and obtain the MTF according to the ESF.

ISO 12233 essentially regards the gray value of an image as the quality of pixels, and the determination of the edge point is to obtain the centroid of each row pixels. This method is simple in calculation and widely used in the MTF measurement of various cameras. However, the influence of noise is not considered. When the imaging noise is large, this method cannot accurately estimate the edge angle, which sometimes causes false detection. Therefore, before using this method for MTF measurement, the actual image needs to be denoised, but the denoising will cause image distortion, which will also bring errors to the estimation of the edge angle.

#### 2.1.2. Hough Transform

Hough transform [23,24] is one of the common methods used for line detection. Any line in the image space can be represented as a point in the parameter space, which is called duality. Hough transform uses this duality to convert the image from the image space to the parameter space for calculation [25]. The dual transformation is shown in Formula (7)
(7)ρ=x⋅cosα+y⋅sinα
where ρ is the polar diameter, α is the polar angle, and (x, y) is the pixel coordinates. For any set of pixels, a corresponding line can be found in the parameter space. Using this method to extract the edge line can greatly shorten the calculation time and improve efficiency, but the detection accuracy is affected by the manual quantization of the parameter space.

### 2.2. LSD Line Detection Method

LSD (Line Segment Detector) was proposed by Rafael Grompone von Gioi in 2012 [26]. Based on Burns’ algorithm [27], this algorithm combines gradient information and direction information of the image to extract straight lines [28,29]. It is mainly divided into three steps as follows:The pixel gradient of the whole image is calculated to generate the corresponding gradient fields, and the pixels whose gradient is within a certain threshold are merged into a region, which is called the line-support region. as shown in Figure 2;Calculate the minimum circumscribed rectangle of each line-support area, and calculate the total number of pixels in the rectangle and the number of pixels whose gradient angle is equal to the angle of the rectangle. Each circumscribed rectangle represents a line to be detected, which is represented by the endpoint, width, center, angle, and length of the rectangle.The Helmholtz principle is used to determine whether each rectangular region can be extracted as a straight line. That is, by comparing with a noise map with a hypothetical independent distribution value of [0,2π], when the number of false alarms $Q_{NFA}$ meets a threshold, then the linear support region is judged to be extractable as a straight line. The formula for $Q_{NFA}$ is as follows:

(8)$$\begin{array}{l} Q_{NFA}={\left(MN\right)}^{5/2}B(n,k,\delta )\\ B(n,k,\delta )=\sum_{j=k}^{n}\left(\frac{n}{k}\right)p^{j}{\left(1-\delta \right)}^{n-j} \end{array}$$
where, M, N is the image size, n is the total number of pixels in the rectangle, k is the total number of pixels in the same direction as the rectangle, and δ is a certain precision. When $Q_{NFA}$ is less than 1, it meets the extraction conditions and can be judged as a straight line.

### 2.3. Algorithm in This Paper—Slanted-Edge Method Based on Improved Zernike Moment

Traditional methods of detecting the edge, such as the difference operation in ISO 12233 and gradient operator in LSD, are sensitive to noise and can only be located at the pixel level. In order to achieve higher accuracy in photogrammetry, there is often a need to locate at the sub-pixel level [30]. At present, the commonly used sub-pixel positioning method mainly uses invariant moments, which are insensitive to noise [31]. The geometric moment, Legendre moment, Hermite moment, and Zernike moment are commonly used. Compared with other moments, the operation is simpler [32]. Fritz Zernike (1888–1966) introduced a group of complex polynomials defined on the unit circle in 1934, which has completeness and orthogonality, so it can represent any square integrable function defined in the unit circle. The Zernike moment was born. Ghosal and Mehrotal first proposed to use Zernike moments to detect sub-pixel edges [33]. In their algorithm, they established an ideal step grayscale model. They calculated four parameters of the model through three different order Zernike moments of the image, and used these four parameters as the basis for judging edges to determine the edges of objects in the image. Li et al. [34] considered the amplification effect of the template and improved the model and the detection accuracy. Ye et al. [35] avoided the error caused by image normalization by extracting Zernike moments first and then normalizing. Lin et al. [36] reduced the complexity of the model operation and improved the detection efficiency by improving the orthogonality.

However, in the actual operation process, different parameter thresholds need to be determined for different images, otherwise, it will cause false detection.

In this paper, considering the problem that the photogrammetry image is complex and the threshold is difficult to determine, based on the Zernike moment sub-pixel edge extraction method, combined with the Otsu algorithm, the threshold is automatically determined, and the sub-pixel position of the edge point is accurately located. The five-point filtering method is used to smooth the ESF sample points and then fit the ESF curve by Fermi function to reduce the influence of noise. The algorithm is described in detail below.

#### 2.3.1. Zernike Moment

Zernike moments for an image f(x,y) is defined as:(9)$$Z_{nm}=\frac{n+1}{\pi }\iint_{x^{2}+y^{2}\leq 1}f(x,y)V_{nm}^{*}(\rho ,\theta )dxdy$$
where, $V_{nm}^{*}(\rho ,\theta )$ is the $V_{nm}(\rho ,\theta )$ conjugate complex, and $V_{nm}(\rho ,\theta )$ is the orthogonal complex polynomial under the definition of polar coordinates; it can be expressed by Equation (10).
(10)$$V_{nm}(\rho ,\theta )=e^{im\theta }\sum_{s=0}^{\frac{\left(n-\left|m\right|\right)}{2}}\infty \frac{{\left(-1\right)}^{s}(n-s)!\rho^{n-2s}}{s!(\frac{n-\left|m\right|}{2}-s)!(\frac{n-\left|m\right|}{2}-s)}$$
where *n* is a positive integer or zero, *m* is a positive or negative integer, n−|m| is an even number, and n≥|m|.

#### 2.3.2. Sub-Pixel Edge Detection Method Based on Zernike Moment

For sub-pixel edge detection based on Zernike moments, the center of the image must be shifted to the coordinate origin, and the image pixel points must be mapped into the unit circle [37]. The Zernike moments for an image f(x,y) in the unit circle are defined as:(11)$$Z_{nm}=\sum_{x}\sum_{y}f(x,y)V_{nm}^{*}(\rho ,\theta )$$

Only three $Z_{nm}$ ($Z_{00}$, $Z_{11}$, $Z_{20}$) of different orders are needed to calculate four positioning parameters (d, φ, h, k) and complete the edge positioning. Firstly, an ideal model for subpixel edge detection is established [38], as shown in Figure 2a, where the circle is the unit circle, the line L is contained by the unit circle, L. The gray values on both sides of L are h and h+k, respectively, d is the vertical distance from the origin to the line L, and φ is the angle between d direction and x axes. Figure 3b shows the result of rotation φ clockwise in Figure 3a.

Zernike moments have rotational invariance, and after rotating the image by φ, the rotated Zernike moment $Z_{nm}^{\prime }$ is:(12)$$Z_{nm}^{\prime }=Z_{nm}e^{\left(-im\varphi \right)}$$

According to Formula (12), when n=2,4, $Z_{n0}^{\prime }=Z_{n0}$; that is, the Zernike moment remains unchanged before and after rotation. When n=1,3, $Z_{n1}$ is a complex number. Since the rotated image is symmetric about the x axis, the imaginary part of $Z_{n1}^{}$ is 0, that is:(13)Im[Zn1′]=sin(φn1)Re[Zn1]−cos(φn1)Im[Zn1]=0

Then φ can be solved:(14)φn1=arctan(Im[Zn1]Re[Zn1])

In conclusion, according to Zernike moments of different orders after rotation, three parameters d, k and h can be deduced.
(15)$$d=\frac{Z_{20}}{Z_{11}^{\prime }}=\frac{Z_{20}}{Z_{11}^{}}e^{-j\varphi }$$
(16)$$k=\frac{3Z_{11}^{\prime }}{2{\left(1-d^{2}\right)}^{2/3}}=\frac{3Z_{11}^{}}{2{\left(1-d^{2}\right)}^{2/3}}e^{j\varphi }$$
(17)$$h=(Z_{00}-\frac{k\pi }{2}+k\arcsin d+kd\sqrt{1-d^{2}})/\pi$$

According to the parameters, the sub-pixel detection formula can be deduced as follows:(18)$$\left[\begin{array}{c} x_{s}\\ y_{s} \end{array}\right]=\left[\begin{array}{c} x\\ y \end{array}\right]+d\left[\begin{array}{c} \cos(\varphi )\\ \sin(\varphi ) \end{array}\right]$$
where, ($x_{s}$, $y_{s}$) is the sub-pixel coordinate of the edge, and (x, y) is the origin coordinate. For digital images, $Z_{00}$, $Z_{11}$, $Z_{20}$ can be obtained through the convolution of images and templates. The larger the template size, the higher the solution accuracy. Set template as N*N. Considering the amplification and correction effect of the template, Equation (18) can be changed as follows:(19)$$\left[\begin{array}{c} x_{s}\\ y_{s} \end{array}\right]=\left[\begin{array}{c} x\\ y \end{array}\right]+\frac{Nd}{2}\left[\begin{array}{c} \cos(\varphi )\\ \sin(\varphi ) \end{array}\right]$$

In this paper, a 7 × 7 template is used [39], and the template is shown in Figure 4.

The parameters are calculated for each pixel of the image, and when the parameters meet the condition $k\geq k_{t}\cap d\leq d_{t}$, the pixel is the edge point, and $k_{t}$, $d_{t}$ are the thresholds. Zernike moment subpixel detection requires accurate selection of thresholds, otherwise it will cause false detection.

#### 2.3.3. Otsu Adaptive Threshold Method

The value of $d_{t}$ has little influence on the result, usually taking $1/\sqrt{2}$, and the value of $k_{t}$ has a great influence on the result. For different images, $k_{t}$ is difficult to determine. In this paper, the Otsu algorithm is used to automatically determine $k_{t}$.

The Otsu algorithm was proposed by OTSU N in 1979 [40]. It is mainly applied to digital image segmentation [41,42], and its main idea is to divide the image into foreground and background parts by using a threshold [43,44,45], to maximize the variance between foreground and background. Suppose the segmentation threshold is t, the proportion of foreground pixels is $\omega_{0}$, the average gray level is $\mu_{0}$, the proportion of background pixels is $\omega_{1}$, the average gray level is $\mu_{1}$, the total average gray level of the image is μ, and the variance between-cluster is g, then:(20)$$g=\omega_{0}\omega_{1}{\left(\mu_{0}\mu_{1}\right)}^{2}$$

The ergodic method is used to maximize g, when g reaches the maximum, the corresponding t is the threshold value.

In this method, the parameter k of each pixel is obtained by traversing the image, k is taken as the calculation object, and the threshold value obtained is $k_{t}$. In order to verify the sub-pixel edge extraction results based on the improved Zernike moment, the edge ROI image (144 × 372) with a noise variance of 0.1 was taken as an example and sub-pixel edge points were extracted; the results are shown in Figure 5. Figure 5b shows that many noise points are detected as edge points due to improper threshold selection in the traditional Zernike moment sub-pixel edge detection results. Figure 5c shows that the improved method can select the threshold value well, the detected edge points are clear and reliable, and the noise points are basically eliminated. After the edge points are accurately extracted, the accurate edge lines can be obtained by least square fitting. The fitted edge straight line is shown in Figure 5d.

### 2.4. MTF Calculation

After the edge line is obtained based on the above method, each pixel of the image is projected along the edge direction to obtain ESF sample points [46]. That is, all the pixels in the ROI are projected to the first line along the edge direction. The gray value of the pixel is the y-axis, and the position of the pixel after projection is the X-axis. The projection method is shown in Figure 6a.

The gray values of pixels were averaged at the same interval after projection as the values of ESF sampling points at this location. The projection interval is generally 4, that is, on average every 1/4 pixel. However, using a fixed integer oversampling ratio will reduce the accuracy and precision of MTF measurement, because there is periodic misalignment between the projection path and the projection interval. Kenichiro [47] proposed a new projection method based on ISO 12233 method, called OMNI-sine method. This method projects the pixels in ROI onto the row coordinate axis perpendicular to the edge line, and adjusts the projection interval according to the edge angle, which improves the accuracy of MTF measurement to a certain extent. The projection mode is shown in Figure 6b, and the projection interval *V* can be defined by Equation (21), where θ  is the edge angle and $\theta_{sym}$ is a symmetric slant angle.
(21)$$\begin{array}{c} V=n\times 2^{\left[\log2^{\left(\sin\theta_{sym}\right)}\right]-\log2^{\left(\sin\theta_{sym}\right)}}\\ \theta_{sym}=\frac{\arccos\left(\cos4\theta \right)}{4} \end{array}$$

In order to avoid the impact of noise, this paper uses the five point filtering method to smooth the ESF sample points; that is, slide a window with a step of 5, take the average value of the elements in the window as the central point element value, use a Fermi function to fit the smoothed ESF sample points into a smooth ESF curve, derive the differential of the ESF curve, and use Gaussian function to fit. The LSF curve is then obtained; after Fourier transformation of LSF, take modulus normalization to obtain the MTF curve. MTF calculation flow is shown in Figure 7.

The MTF curve can obtain MTF value at each frequency, which is usually represented by MTF value at Nyquist frequency (half of discrete signal sampling frequency [48]). The larger the MTF value is, the smaller the difference before and after imaging is, which means the smaller the change in light energy distribution is, the better the imaging performance of the optical system is. If the MTF of an optical system is 1, then the distribution of light energy of the system has not changed at all, and the performance is very good. However, in practical applications, the MTF of optical systems is generally less than 0.4.

## 3. Experimental Procedure and Results

In order to verify the accuracy of the algorithm in this paper for edge angle detection and MTF calculation under different conditions, a simulation experiment is designed. First, an ideal edge image is generated by a computer, and then different degrees of noise and blur are added to the ideal edge image to obtain a simulation image, using the ISO 12233 method, Hough transform method, LSD method and the algorithm proposed in this paper to perform edge detection on these images (OMNI-sine method does not participate in edge angle detection experiment comparison because it uses the same edge detection method as ISO 12233), according to the deviation of the detected edge angle and the theoretical edge angle to verify the accuracy of edge detection. At the same time, the deviation of angle detection is verified by an outdoor calibration field radiation target experiment. The accuracy of the four algorithms is analyzed, along with the characteristics and regularity of the four algorithms. The system MTF is simulated by the Gaussian kernel function, and the MTF calculated by the five methods (ISO 12233, OMNI-sine, Hough, LSD and Algorithm in this paper) is compared to verify the accuracy of the algorithm. The experimental process is shown in Figure 8. The experimental environment is a PC with a memory of 16GB, and the simulation experiment was carried out through MATLAB 2020a software produced by MathWorks Company in Natick, MA, USA.

### 3.1. Simulation Experiment

#### 3.1.1. Edge Angle Comparison

The ideal black and white edge image of 400 × 400 is generated by a computer, the edge angle is 8°, the gray value of the high reflectivity area is 255, and the gray value of the low reflectivity area is 0. The size of the captured ROI image is 144 × 372, as shown in Figure 9. Add Gaussian noise, which is divided into 15 levels according to variance: 0, 0.002, 0.004, 0.006, 0.008, 0.01, 0.012, 0.014, 0.016, 0.018, 0.02, 0.04, 0.06, 0.08, 0.1. The blur is mainly performed by convolution operation with a point spread function image of known size and the ideal edge image. The larger the point spread function image, the stronger the blurring degree. The fuzzy degree in this paper is divided into 11 levels. The size of the point spread function is set to N × N, and the values of N are taken as 0, 2, 4, 6, 8, 10, 12, 14, 16, 18, and 20, respectively. A total of 165 sets of pictures are used.

ISO 12233, Hough transform method, LSD method and the algorithm proposed in this paper were used to detect the edge of 165 groups of simulated images, record the edge angle detected in each experiment, and measure the difference between it and the theoretical edge angle of 8°. The mean and variance of edge angle deviation were detected by each method. The smaller the mean, the higher the detection accuracy and the smaller the variance, indicating that the method is more stable under the experimental conditions. The results are shown in Table 1.

It can be seen from Table 1, the mean angle deviation by the algorithm proposed in this paper is the smallest, only 0.032°, and the detection accuracy is the highest. The average value of LSD detection is the largest, reaching 0.246°; comparing the variance in edge angle deviations detected by each algorithm, the variance in the Hough transform is the smallest, only 0.03°, indicating that this method is the most stable under experimental conditions. However, the variance of LSD detection reaches 1.215°, indicating that it is relatively unstable under the experimental conditions of this paper.

#### 3.1.2. Comparison of MTF Test Results

The output of the optical imaging system is the result of the convolution of the input with its impulse response function. Therefore, if the response function of the target scene and the system is known, the output of the system can be obtained by convolution calculation, that is, the imaging result of the optical imaging system. At the same time, if the impulse response function is known, the modulation transfer function of the optical imaging system is also known. The edge method uses the output of the optical imaging system with a step response to calculate the MTF of the system so that the calculated value can be compared with the system value obtained directly using the impulse response function, to confirm the accuracy of the algorithm. The specific method is as follows: firstly, filter the ideal edge image with a low-pass filter with known parameters to obtain the fuzzy edge image and the MTF of the system, then obtain the calculated MTF through the method described above and compare them with the respective system MTF.

In this paper, the Gaussian kernel function is used as the low-pass filter, and the width parameter σ is taken as 0.6. The detection results are shown in Figure 10.

The system MTF value at the Nyquist frequency is 0.1586, and the comparison between the MTF at the Nyquist frequency and the system value of each method is shown in Table 2.

It can be seen that the MTF value measurement by the algorithm in this paper has the smallest deviation from the system value, only 0.0392, which is closer to the real result.

### 3.2. Outdoor Calibration Field Experiment

In order to further verify the accuracy of the algorithm in this paper in the actual situation, four methods are used to detect the edge angle by taking photos of the radiation target in the outdoor calibration field. The outdoor calibration field is located on the Chaoyang Campus of Jilin University. The radiation targets are designed with 10 black and white target bars, each with a central angle of 5°, a total central angle of 100° and a radius of 3.5 m. The DJI Phantom 4 Advanced UAV was used to obtain aerial photos, with a flight height of 50 m, a lens focal length of 8.8 mm, and a pixel size of 0.00241 mm and the photo was taken at 50 m altitude. Two groups of ROIs are intercepted in the radiation target. Each group was divided into 8 images (each image with only 1 edge) from left to right, and four methods were used to detect the edge angle of each image; furthermore, the difference between the edge angles detected in two adjacent images was calculated and compared with the theoretical value of 5 degrees, the difference between the edge angle of one picture apart was calculated and compared with the theoretical value of 10 degrees, and the difference between the edge angle of two picture apart was calculated and compared with the theoretical value of 15 degrees, The radiation target is shown in Figure 11. Then, 18 comparison values were obtained for each method in each group, and the mean square error of each method was calculated; see Table 3.

It can be seen from Table 4 that in the two groups of 90 controlled experiments, the MSE of the edge angle of the algorithm in this paper is the smallest among the methods in the same group, indicating that the results of the algorithm in this paper are more accurate. The edge angle has some influence on the calculation of MTF, and in general, the edge angle is taken as 2°–10°. Li [24] confirmed through extensive experiments that the best results are obtained when the edge angle is taken as 2.5°–6°. In this paper, two images (the first and third images) within this range are selected separately from two sets of radiation target images to calculate MTF and take the average value. The results are shown in Figure 12 and Table 4.

It can be seen from Table 4 that the MTF measured by each group of ROI is relatively close; the overall average MTF is 0.1225 (cy/pix). The MSE values of the two groups of MTF are only 0.0115 and 0.0133, and the overall MSE is 0.0121, with high accuracy, which proves that the algorithm in this paper can be well applied to MTF measurement.

## 4. Discussion

### 4.1. The Noise and Blur Tolerance Analysis of the Algorithm in This Paper

In order to analyze the tolerance of four methods (ISO 12233, Hough, LSD and Algorithm in this paper) to noise and blur and avoid the influence of certain singular values on the overall results, the deviation between the edge angle detection result and the ideal edge angle (8°) in the simulation experiment are divided into five levels, and the deviation within 0–0.05° is level 1, which is marked in green. Level 2 is 0.05–0.1°, marked with blue; 0.1–0.15° is level 3, marked with yellow; 0.15–0.3° is level 4, marked with red; finally, 0.3° or more is level 5, and is marked with black. At the same time, taking the noise condition as the x axis, the blur condition as the y axis, and each level representing a unit, a two-dimensional coordinate system was established. Using the data at level 1 as the base and the data at other levels adjacent to it as the demarcation, the demarcation line for the best detection result is obtained. The results are shown in Figure 13.

The distribution of edge angle deviation detected by different methods at each level is statistically analyzed, and the results are shown in Table 5.

As can be seen from Table 5, the amount of data with the detection result of the method in this paper at level 1 is the largest, reaching 152, accounting for 92.1% of the total experimental data, which proves that the detection effect of the method in this paper is the best.

Each demarcation line is shown in Formula (22):(22)$$\begin{array}{l} L1:y=-7000x+42\;\;x\in (0,0.006),y\in (0,20)\\ L2:y=-2000x+22\;\;x\in (0,0.008),y\in (6,20)\\ L3:y=-2000x+44\;\;x\in (0,0.022),y\in (0,20)\\ L4:y=-500x+46\;\;x\in (0,0.092),y\in (0,20) \end{array}$$

It can be seen from Figure 13 that the four methods can achieve better results within their respective demarcation line equations. The horizontal and vertical axis intercepts of the demarcation line equation can represent the tolerance of noise and blur. The intercept of the demarcation line equation L4 of the algorithm in this paper is the largest, indicating that the tolerance of noise and blur is better than that of the other three methods.

### 4.2. Limitations of the Algorithm in This Paper

The algorithm proposed in this paper also has some limitations.

First of all, although the algorithm proposed in this paper can locate edges accurately, the detection accuracy is affected by the Zernike moment template. The larger the template, the higher the detection accuracy and the more complicated the calculation. The running time of each method is shown in Table 6.

Comparing the running time of several methods in this paper, it can be found that compared with the traditional ISO 12233 method, the running time of the OMNI-sine method is similar to that of the ISO 12233 method, the Hough transform shortens the running time by 7% to a certain extent; the LSD method needs to search pixel by pixel to generate a line support area, so the running time is the longest, increased by 38.8%; the running time of the algorithm in this paper increased by 19.4%. If the edge area can be roughly positioned first, and then the Zernike moment can be used for sub-pixel accurate correction, the operation time can be shortened and the work efficiency can be improved.

Secondly, since LSF is obtained by derivation of ESF, some small errors of ESF will also be amplified to MTF. Nowadays, the smoothing methods for ESF are generally di-vided into parametric methods and nonparametric methods. Parametric methods include polynomial fitting, Fermi function fitting, etc. The nonparametric method includes smooth filtering, etc. However, different methods can only work well for some specific sensors. Therefore, it is also worth exploring to analyze and compare these methods and find the most universal smoothing method.

Finally, the aerial camera will have image shift during operation, which will lead to a decline in image quality and affect the MTF measurement. This study did not consider the impact of image motion on MTF. Improvement of the image motion compensation model of MTF is also the focus of future research. 

## 5. Conclusions

In this paper, a new method for measuring MTF based on the improved Zernike moment is proposed. First, the ROI is extracted from the original image. After the threshold value of the Zernike moment is automatically selected by the Otsu algorithm, the edge points are detected and better eliminate the impact of noise points. The least squares method is used to fit the edge line, and the image is projected along the edge line to obtain the ESF sample. After smoothing using the five-point filtering method, the ESF curve is fitted with the Fermi function, and the LSF curve is obtained by derivation of the ESF. Finally, Fourier transform and normalization are performed to obtain the MTF curve. The edge angle is verified by computer simulation images and outdoor calibration field experiments, and MTF verification is carried out by Gaussian low-pass filter simulation system MTF, and compared with the ISO 12233 method, OMNI-sine method, Hough transform method and LSD method. The conclusions are as follows:The algorithm in this paper detects the edge angle with higher accuracy

Comparing the edge angle deviation detected by the simulated images, the detection accuracy of ISO 12233 and Hough transform are close, with average values of 0.059° and 0.050° respectively; the average value of LSD detection is the largest, reaching 0.246°, mainly due to the large noise and blur, the detection angle deviation is large; the average value of the algorithm in this paper is the smallest, only 0.032°, and the detection accuracy is the highest.

Verified by the radiation target in the outdoor calibration field, the edge angle detected by the algorithm in this paper has the smallest MSE and more accurate detection results than other methods.

The algorithm in this paper has a better tolerance of noise and blur

Through the classification of the experimental data, it is found that the data with the ISO 12233 method at level 1 accounted for 60.6% of the total experimental data; the data with the Hough transform detection results at levels 1 and 2 accounted for 47.3% and 49.1% of the total experimental data respectively, which are relatively stable. The detection results of the LSD method account for 86.1% of the total experimental data, but the data at level 5 is the most numerous, accounting for 3.6% of the total experimental data, and the data at this level have a large deviation, resulting in the mean and variance of this method being the largest among the four methods. The method in this paper has the largest number of detection results at level 1, accounting for 92.1% of the total experimental data, and the tolerance to noise and blur is the best. For the demarcation line equation solved by each method, the intercept of the demarcation line equation of the algorithm in this paper is the largest, which further confirms that the method in this paper has better tolerance of noise and blur than other methods.

The accuracy of MTF measurement by the algorithm in this paper has been improved

By simulating the system MTF value through Gaussian kernel function and comparing the difference between MTF and system value at Nyquist frequency of five methods, we found that the deviation of MTF measurement by the algorithm in this paper is the smallest, only 0.0392. The MTF measurement by the algorithm in this paper is closest to the real value.

In the subsequent work, we will further explore the influence of the ESF smoothing method on MTF and improve the motion compensation model of MTF to establish a set of MTF measurement methods completely applicable to aerospace sensors.

## Figures and Tables

**Figure 1 sensors-23-00509-f001:**
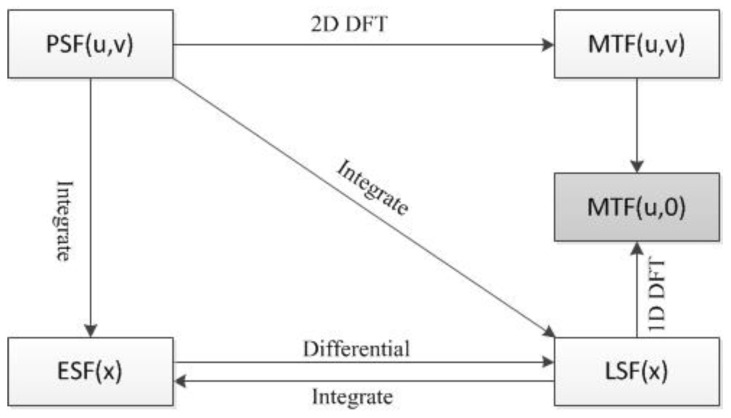
Relationship between PSF, LSF, ESF and MTF.

**Figure 2 sensors-23-00509-f002:**
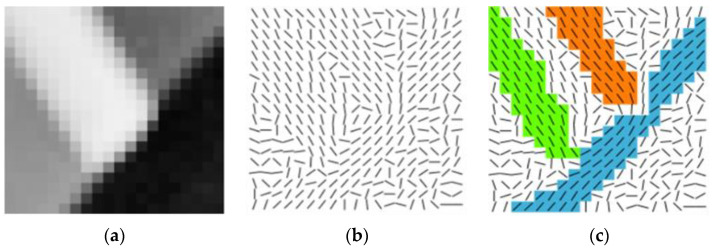
Gradient fields and line-support region of LSD Algorithm: (**a**) Original image; (**b**) Gradient fields; (**c**) Line-support region.

**Figure 3 sensors-23-00509-f003:**
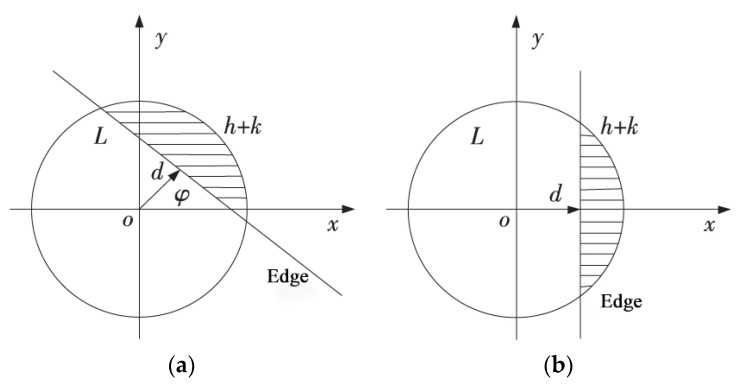
Ideal model for sub-pixel edge detection: (**a**) Original edge image; (**b**) Edge image after rotation.

**Figure 4 sensors-23-00509-f004:**
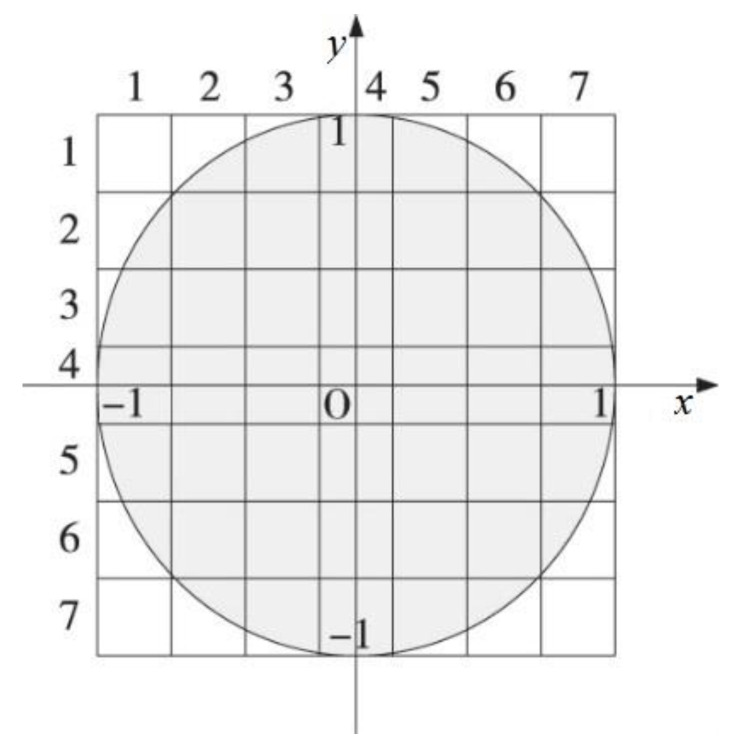
7 × 7 Template.

**Figure 5 sensors-23-00509-f005:**
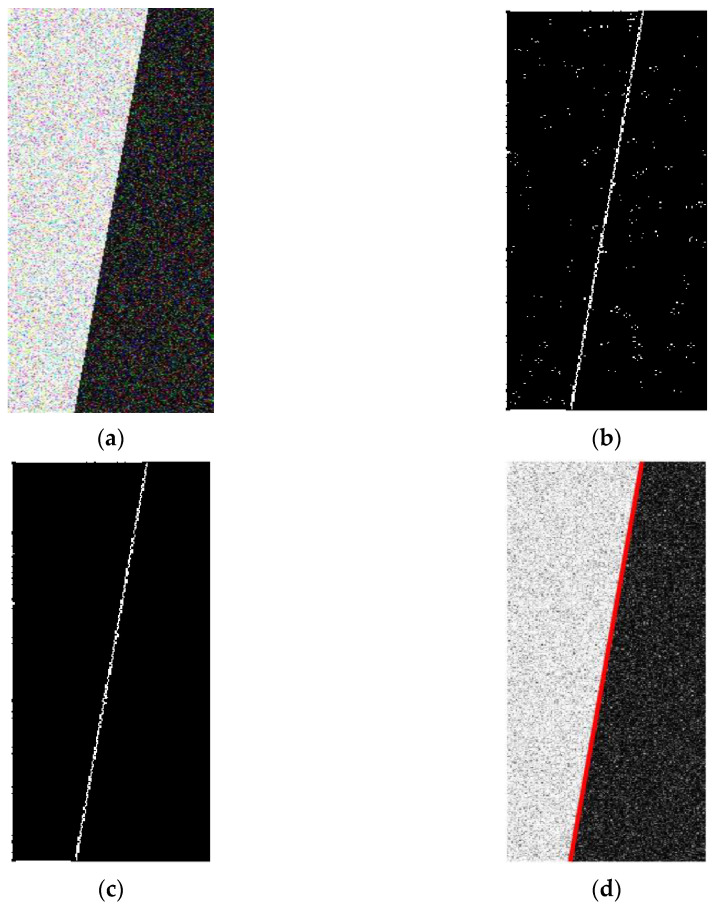
Sub-pixel edge detection results: (**a**) Edge image with noise variance of 0.1; (**b**) Sub-pixel edge detection results by traditional Zernike moments; (**c**) Sub-pixel edge detection results by Improved Zernike moment; (**d**) Detected edge straight line.

**Figure 6 sensors-23-00509-f006:**
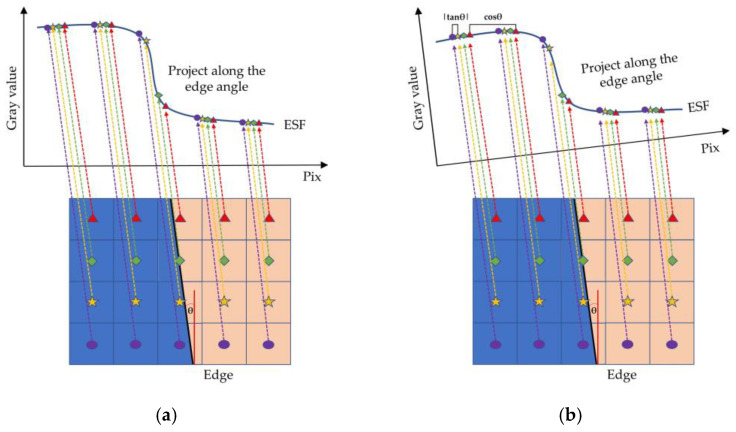
Projection Diagram. (**a**) Projection Diagram of ISO 12233, Hough, LSD, and Algorithm in this paper. (**b**) Projection Diagram of OMNI-sine.

**Figure 7 sensors-23-00509-f007:**
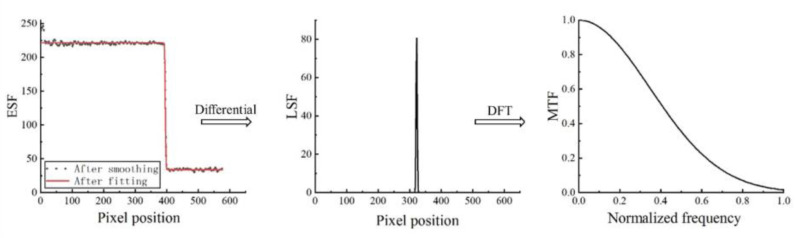
MTF Calculation Process.

**Figure 8 sensors-23-00509-f008:**
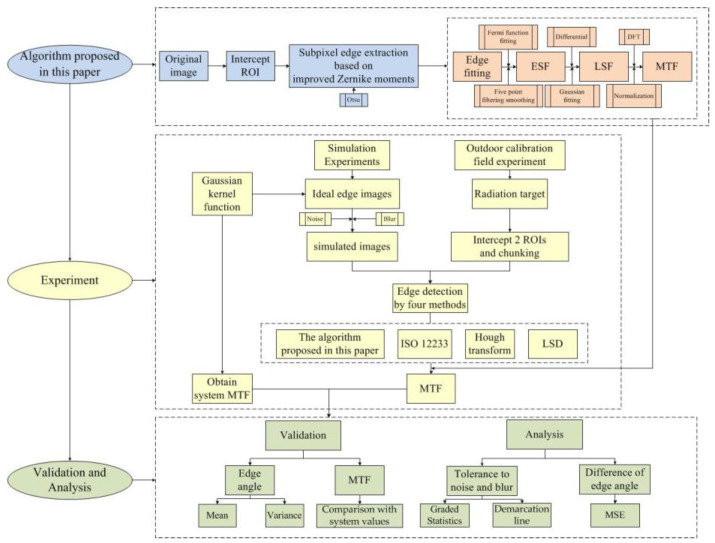
Experimental flow chart.

**Figure 9 sensors-23-00509-f009:**
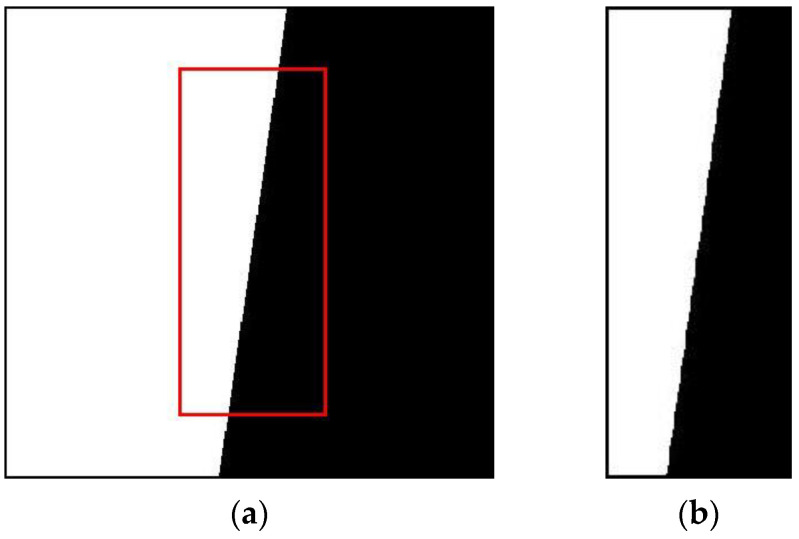
Ideal edge image and ROI: (**a**)the ideal edge image; (**b**)the ROI intercepted from Figure 9a.

**Figure 10 sensors-23-00509-f010:**
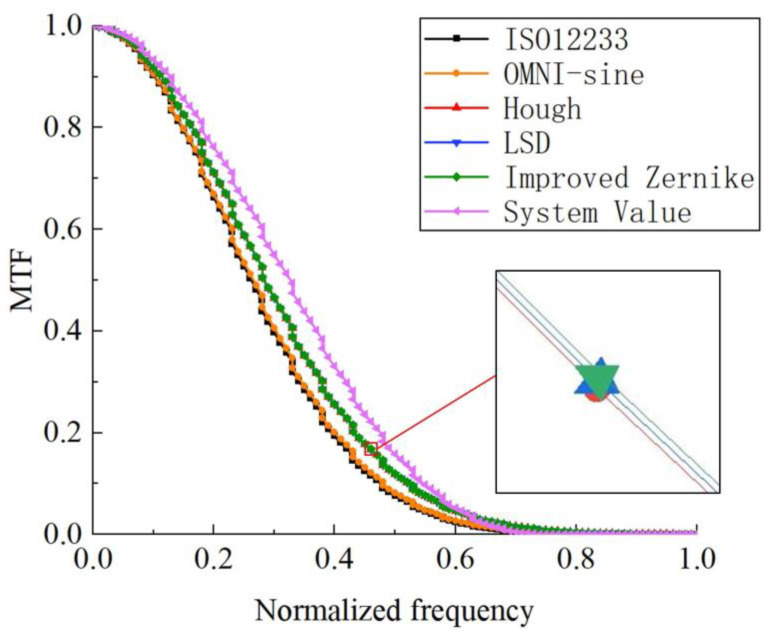
MTF of each method compared with the system value.

**Figure 11 sensors-23-00509-f011:**
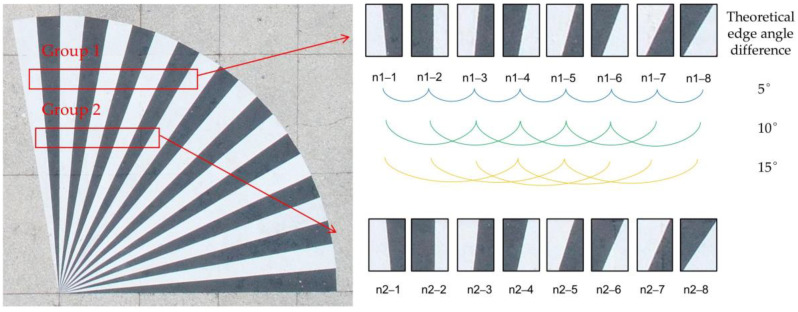
Experimental ROI.

**Figure 12 sensors-23-00509-f012:**
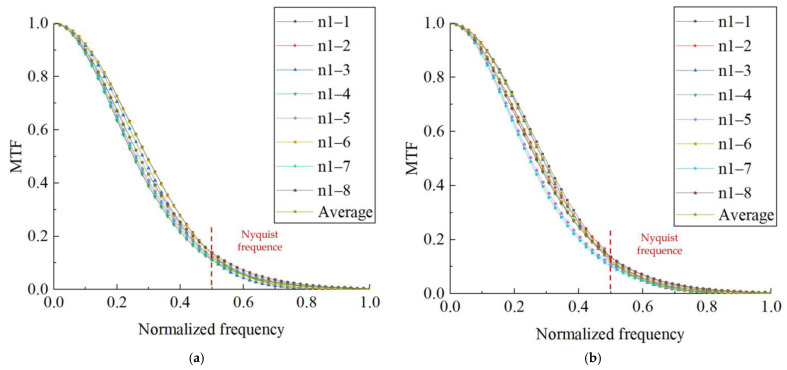
MTF calculated from the radiation target: (**a**) MTF calculated by the proposed algorithm in this paper for group 1; (**b**) MTF calculated by the proposed algorithm in this paper for group 2.

**Figure 13 sensors-23-00509-f013:**
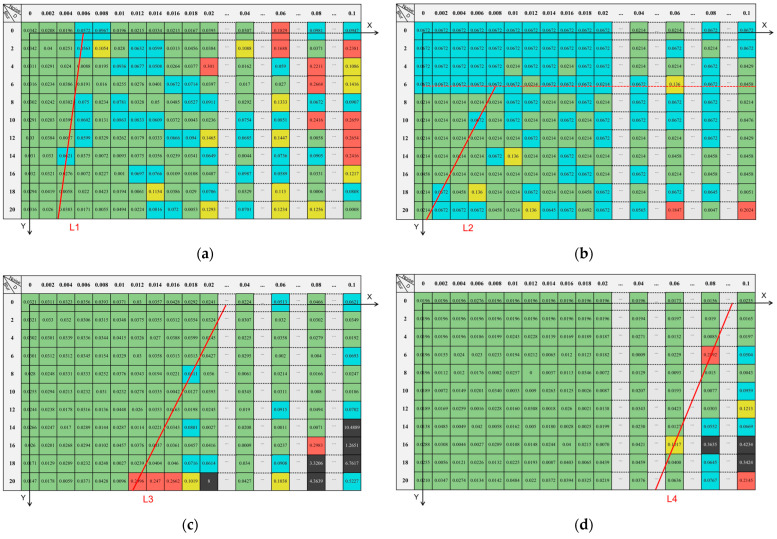
The results and demarcation line of each method: (**a**) detection results and demarcation line L1 of ISO 12233; (**b**) detection results and demarcation line L2 of Hough transform(The red dotted line is the value range of the demarcation line); (**c**) detection results and demarcation line L3 of LSD; (**d**) detection results and demarcation line L4 of this algorithm.

**Table 1 sensors-23-00509-t001:** Mean and variance of the edge angle deviation of the test results by different method.

Indicator	Method			
	ISO 12233	Hough Transform	LSD	Algorithm in This Paper
Mean angle deviation (°)	0.059	0.050	0.246	0.032
Angle deviation variance (°)	0.057	0.030	1.215	0.056

**Table 2 sensors-23-00509-t002:** MTF comparison at Nyquist frequency.

Indicator	MTF and Its Deviation (cy/pix)				
	ISO 12233	OMNI-Sine	Hough Transform	LSD	Algorithm in This Paper
Measurement value of MTF	0.0769	0.0815	0.1176	0.1184	0.1194
Deviation from system value	0.0817	0.0771	0.0410	0.0402	0.0392

**Table 3 sensors-23-00509-t003:** MSE of experimental results.

Group No.	MSE (°)			
	ISO 12233	Hough Transform	LSD	Algorithm in This Paper
1	0.460	0.396	0.256	0.216
2	0.660	0.406	0.340	0.310

**Table 4 sensors-23-00509-t004:** MTF calculated from the radiation target at Nyquist frequency.

No.	MTF (cy/pix)		
	Group 1	Group 2	Total
1	0.1364	0.1366	-
2	0.1125	0.1167	-
3	0.1131	0.1119	-
4	0.1106	0.1333	-
5	0.1239	0.1099	-
6	0.1333	0.1263	-
7	0.1195	0.1012	-
8	0.1394	0.1354	-
Average	0.1236	0.1214	0.1225
MSE	0.0115	0.0133	0.0121

**Table 5 sensors-23-00509-t005:** Distribution and proportion of the grade of the test results by different methods.

Quantity (Percentage)	Method			
	ISO 12233	Hough Transform	LSD	Algorithm in This Paper
Level 1	100 (60.6%)	78 (47.3%)	142 (86.1%)	152 (92.1%)
Level 2	42 (25.4%)	81 (49.1%)	11 (66.7%)	6 (3.6%)
Level 3	13 (7.9%)	4 (2.4%)	2 (1.2%)	2 (1.2%)
Level 4	10 (6.1%)	2 (1.2%)	4 (2.4%)	2 (1.2%)
Level 5	0 (0%)	0 (0%)	6 (3.6%)	3 (1.8%)

**Table 6 sensors-23-00509-t006:** Running time of different methods.

Method	ISO 12233	OMNI-Sine	Hough Transform	LSD	Algorithm in This Paper
Consumed time (s)	2.78	2.89	2.58	3.86	3.32

## Data Availability

Not applicable.

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
