# Peer review of "MTF Measurement by Slanted-Edge Method Based on Improved Zernike Moments"

_sensors, 2023, doi:10.3390/s23010509_

Round 1
Reviewer 1 Report
The manuscript by Zhang, Wang, Wu, and Gao addresses the application of so-called improved Zernike moments to edge detection for MTF estimation using the slanted edge method.
The work contains both simulations and experimental results. The simulations appear to show that the estimation accuracy and standard deviation are comparable to or better than those of a method called LSD and better than the ISO-recommended standard method and the Hough transform method. The parameter selection (dt and kt) in the Zernike method appears to have been carefully investigated. Therefore, consideration for publication in an archival scientific journal makes sense.
I did not find the manuscript particularly clear, and I found it obscure in places and lacking some information. Therefore, I cannot recommend publication at this stage. Here are my comments
1) There is no information about the optical system being tested or about the sensor.
2) The method recommended is called “improved Zernike moments”. It seems that the improved Zernike moments were introduced no later than 2002 by Ye and Peng, DOI 10.1088/1464-4258/4/6/304. Perhaps that publication, or any earlier publication on the subject, should be cited? Or is this “improved Zernike moments” method different from former methods by the same name?
3) I failed to understand why exactly the method is called “improved”, is it because of the normalization?
4) The Zernike moments method involves a radius, measured as a specified number of pixels. Is that a relevant parameter for the method performance?
5) A noise advantage is claimed. However, only Gaussian noise has been tested in the simulation and not much on noise is said in the experimental part. Is this relevant or not? Also, nothing is said about possible other existing methods aside from the ISO standard, Hough and LSE: are there any or not?
6) Some experimental or simulation descriptions that would be hard to reproduce do not seem to add much value to the publication. While section 3.1 does contain the most convincing results, I feel that it could easily be shortened by approximately 50% without harm.
7) The title is unclear: do you detect an MTF, or do you measure an MTF? The detection applies to the existence of an edge, then the exact position of the edge should be estimated (which is when the improved Zernike moments come into play) and then the MTF can be estimated, leading to an MTF measurement. May I suggest “use of improved Zernike moments to estimate edge location for MTF measurement”?
8) Line 179: please explain what is meant by the “Helmholtz principle” and why it is relevant here.
9) Line 284: what do you mean by “along the edge angle”?
10) The introduction is lengthy, and there are many references of little interest to this work, while other references of direct interest (see above) appear to be missing.
Reviewer 2 Report
In this paper, MTF detection by slanted-edge method has been proposed. There are fundamental problems that authors should consider as follow:
1. The compared baselines is classic, but they are somewhat outdated. There are many newly published works on evaluation of optical imaging in the years of 2021, and 2022. You should compare the proposed method against more recent SOTA baselines.
2. The tables should be analyzed more qualitatively in the Results and Discussion section.
3. Outdoor Calibration Field Experiment needs to be enriched.
4. Discussion section needs to be enriched.
5. The abstract should be rewritten.
6. In the introduction, the innovation of the manuscript is well discussed, but it is better to add the related research section to the introduction.
7. What have been the limitations of your work?
8. The conclusion is brief. Suggestions for future works should be added in this section.
Round 2
Reviewer 1 Report
The authors have taken the reviewers' comments into consideration to a fair extent. I think that the manuscript can now be accepted for publication in Sensors.
Reviewer 2 Report
I think the following points need be more demonstrated from other perspectives, but the author's reply is not very satisfactory.
1. The compared baselines is classic, but they are somewhat outdated. There are many newly published works on evaluation of optical imaging in the years of 2021, and 2022. You should compare the proposed method against more recent SOTA baselines.
Authors: “After consulting the literature, we did not find a suitable new method for MTF measurement by slanted-edge method from 2021 to 2022.” I think any MTF measurement can compare the results, not necessarily based on the hypotenuse method, so there should be relevant literature methods for comparative analysis after 2012.
3. Outdoor Calibration Field Experiment needs to be enriched Outdoor Calibration Field Experiment is the real application scenario, so, can multiple groups verify the accuracy of the analysis method in different situations?
4. Discussion section needs to be enriched. Discussion and the Result are different. In this revision, the author only adjusted the figure and content of the result part to the Discussion part, which is inappropriate.
Author Response
请参阅附件
